# Adaptation of Ancient Techniques to Recreate 'Wines' and 'Beverages' Using Withered Grapes of Muscat of Alexandria

**Mkrtich Harutyunyan** [1] , **Renato Viana** [1], **Joana Granja-Soares** [2], **Miguel Martins** [2], **Henrique Ribeiro** [1] and **Manuel Malfeito-Ferreira** [1,*]

[1] Linking Landscape, Environment, Agriculture and Food Research Center (LEAF), Associated Laboratory TERRA, Instituto Superior de Agronomia, Universidade de Lisboa, Tapada da Ajuda, 1349-017 Lisboa, Portugal; mharutyunyan@isa.ulisboa.pt (M.H.); renato.mem@gmail.com (R.V.); henriqueribe@isa.ulisboa.pt (H.R.)

[2] Departamento de Ciências e Engenharia de Biossistemas (DCEB), Instituto Superior de Agronomia, Universidade de Lisboa, Tapada da Ajuda, 1349-017 Lisboa, Portugal; joanagranjasoares@gmail.com (J.G.-S.); mmartins@isa.ulisboa.pt (M.M.)

* Correspondence: mmalfeito@isa.ulisboa.pt

**Abstract:** The production of wines using ancient techniques is a present trend with commercial interest among consumers valorising their historical background. Therefore, the objective of the present work was to recreate wines and *piquettes* produced from dehydrated grapes, vinified according to the techniques described in classical Roman agricultural manuals. Muscat of Alexandria grapes were harvested and subjected to greenhouse drying under ambient temperature for 7 days, during the 2020 and 2021 harvests. When weight loss was approximately 30%, grapes were processed according to different protocols, including a rehydration step using saltwater or white wine (2020 harvest). Fermentation was conducted with the addition of commercial yeast without sulphur dioxide supplementation. The *piquettes* were obtained from the pressed pomaces diluted with water solution (5 g/L tartaric acid). The 2020 wines showed alcoholic content and residual sugar ranging from 14.8 to 17.0% (*v/v*), and 0.8 g/L to 18 g/L, respectively. Volatile acidity was less than 1 g/L (as acetic acid) in all wines, except for the fermentation of crushed grapes alone, which yielded 2.3 g/L volatile acidity. The fermentation of dehydrated crushed grapes in the semi-industrial trial run in the harvest of 2021 yielded 1.1 g/L volatile acidity. The *piquettes* analysis showed ethanol ranging from 10.2% (*v/v*) to 16.0% (*v/v*), reducing substances less than 2 g/L and volatile acidity less than 0.8 g/L. Overall, the physicochemical analysis showed that it was possible to recreate ancient winemaking techniques that may be further improved to produce commercially and legally acceptable wines.

**Keywords:** dehydrated grapes; Muscat of Alexandria; sweet wines; *passito* wines; Roman wines; mineral content; *piquette*; *água-pé*

## 1. Introduction

The vinification of dehydrated grapes has been known since at least 1600 BC in Ancient Anatolia [1]. This ancestral technique became popular throughout the Greco-Roman classical world [2,3]. Natural sun-drying, whether on-vine or off-vine, is still considered the most traditional and commonly used method for withering grapes in many Mediterranean countries [4]. The wines are commonly referred to as *passito*-style and normally have low ethanol content (around 11–13%) and high residual sugar [5]. Nevertheless, these types of wines are also characterized by a high content of ethanol, frequently attaining about 15 to 17% (*v/v*) [6,7].

In Classical Rome, there was another type of based wine beverage—*lora* or *vinum operarium*—made after the addition of water to the grape pomace (i.e., the skins, seeds and stems) after several pressings [8]. This type of beverage is characterized by a low-alcohol content (ranging from 4% to 8%, *v/v*) and has extremely limited aging potential [9]. The

actual descendants of such beverages are known in Portugal as *Água-pé*, in France as *Piquette* [10] and in Italy as *Acqua pazza* [11]. The European Union defines this product not as wine, but as a beverage obtained exclusively from the fermentation of grape pomace macerated or leached with water, limiting its commercialization [12].

The variety Muscat of Alexandria (*Vitis vinifera* L.) is among the most common for sweet wine production in the Mediterranean basin [13]. Classical Roman authors describe several techniques of vinifying dried grapes alone or by mixing with seawater or wine [2]. In particular, the translated agricultural treatises of Columella (*c.* 1st century AD) and Pliny the Elder (*c.* 23/24–79 AD) are of great importance, providing a detailed description of several techniques that are presently not commonly explored [14,15].

Limited research has been conducted to describe ancient wine-making techniques and wine styles from an experimental and technical perspective [16–18]. Recently, in the context of new wine market trends, there has been a demand for products with a strong historical and cultural attachment [19], among which stand out wines obtained by different grape drying methods and the *piquettes*. Therefore, the objective of the present work was to adapt the ancient Roman techniques to present technological conditions and evaluate their potential for the production of *passito* and *piquette*-style beverages.

## 2. Materials and Methods

### 2.1. Sample Collection and Dehydration Process

Muscat of Alexandria grapes were grown in the experimental vineyard of Instituto Superior de Agronomia (Lisbon, Portugal) at 38°42′27.5″ N latitude, 9°10′56.3″ W longitude and a low altitude of approximately 50 m above sea level. All production, subjected to the same viticultural practices, was directed to the trials during the 2020 and 2021 harvest years. When the grapes reached approximately 25 °Brix, they were manually harvested and transported to the greenhouse and were totally covered by double-polyethylene film. Bunches were placed on stainless steel nets at a height of 60 cm above the ground and on the ground in air-permeable black plastic baskets. During the drying period, rotten berries were removed and bunches were turned over early every morning at the same time, thus minimizing different drying rates. Duplicate samples of 100 randomly selected berries were taken to determine either ripening or drying progress. After 7 days of dehydration, grapes were processed when they reached approximately 30 °Brix and processed according to different protocols, during the 2020 and 2021 harvests.

### 2.2. Grape Processing

During the 2020 harvest, withered grapes were subjected to six different treatments (Table 1). The rehydration was performed by submerging the berries with saltwater solution (a well-known practice used by ancient Greeks and Romans) [8,20] or with one-year-old wine in plastic boxes for 48 h at room temperature. The basic steps of the winemaking process are illustrated in Figure 1. The boiled juice (0.6 L) was obtained by heating juice (1.6 L) from hand crushed berries (3.44 kg) in a 4-L borosilicate glass beaker, corresponding to the volume reduction of the Roman *sapa* [2]. In the 2021 harvest, the wine was produced on a semi-industrial scale (600 kg of fresh grapes) following the CW protocol.

### 2.3. Fermentation

During the 2020 harvest, the juices of withered grapes were fermented in 20-L glass demijohns with the addition of commercial *Saccharomyces cerevisiae* at a rate of 30 g/hL (Fermol® Mediterranée, AEB SpA, Brescia, Italy). Fermentation was followed until dryness or constant density. Upon completion of fermentation, the wines were transferred into 5-L demijohns after removal of suspended solids. Potassium metabisulfite (Merck, Darmstadt, Germany) was directly added to the wines to obtain approximately 40 mg/L of free sulphur dioxide.

**Table 1.** Summarised description of the different grape processing modalities based on Columella [14] and Pliny [15].

| Wine Products | Code | Withered Grapes (kg) | Description |
|---|---|---|---|
| Obtained from withered grapes | PW | 20.0 | Free-run juice from grape trodden by foot (*prototropum*) |
| | SW | 33.8 | Juice from rehydration of grapes with 15 L saltwater (15 g/L NaCl) |
| | StW | 22.6 | Juice from rehydration of grapes with 5 L of one-year-old white wine [a] |
| | WW | 22.6 | Juice from rehydration of destemmed grapes with 5 L of one-year-old white wine |
| | CW | 33.8 | Juice of destemmed grapes after pressing |
| | CWC | - | CW wine added of grape juice boiled down to one-third (*sapa*) |
| | CW21 [b] | 270 | Juice of destemmed grapes after pressing |
| | CW21L | - | Wine resulting from the gross juice lees by settling for 24 h under room temperature |
| *Piquettes* from grape pomaces [c] | PSW | 24.5 | Pomace from SW wine plus 15 L of tartaric acid solution |
| | PStW | 34.2 | Pomace from StW wine plus 20 L of tartaric acid solution |
| | PWW | 23.0 | Pomace from WW wine plus 15 L of tartaric acid solution |
| | PCW | 17.8 | Pomace from CW wine plus 20 L of tartaric acid solution |

[a] White wine from the vintage of 2019 with 12.5% (*v*/*v*) ethanol, 8.1 g/L total acidity, 0.2 g/L volatile acidity, 0.5 g/L reducing substances, 115 mg/L total sulphur dioxide. [b] Harvest of 2021. [c] Weight of pomace diluted with a solution of 5 g/L tartaric acid.

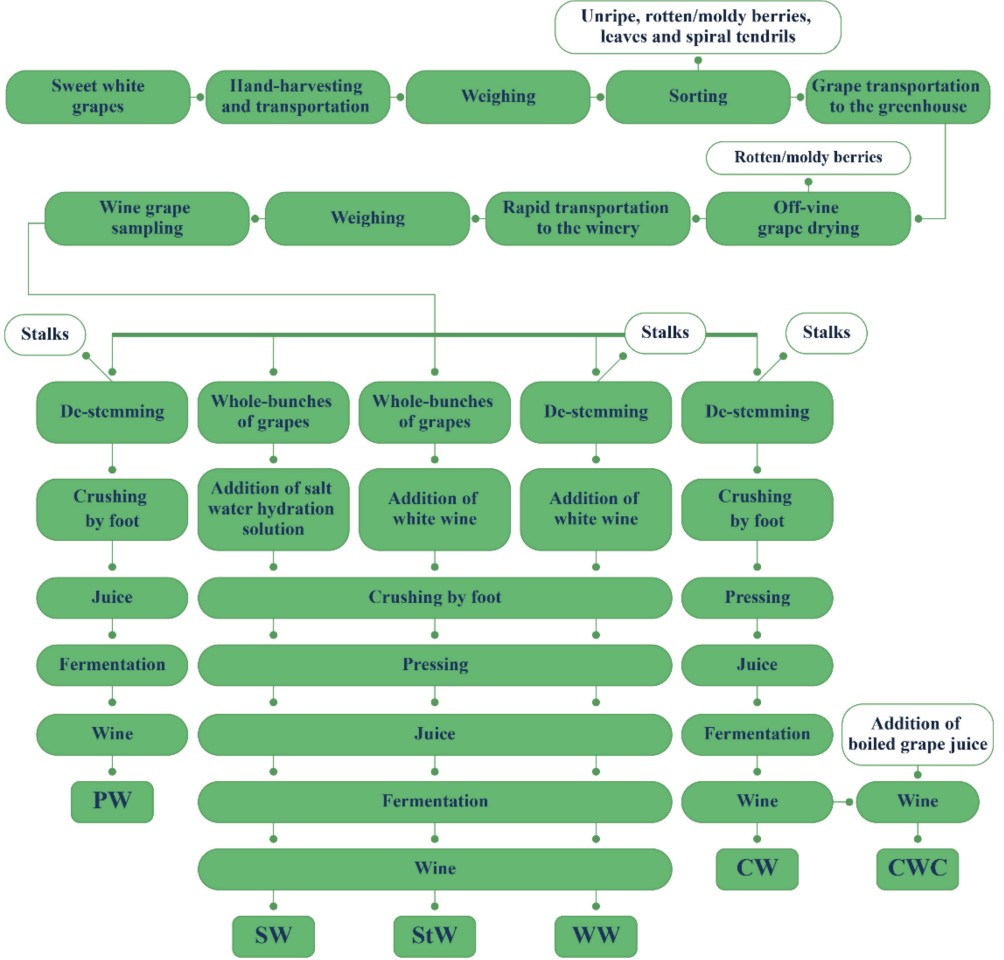

**Figure 1.** The simplified flow diagram of the withering and winemaking processes.

During the 2021 harvest, the fermentation was carried out individually in two large open-top 100-L stainless-steel cylindrical tanks (Fratelli Marchisio & C. S.p.A., Imperia, Italy) at an average temperature of 24 ± 1 °C. The juice was added to 80 mg/L potassium metabisulphite and 30 g/hL Fermol starter. The following steps were as described for the CW sample of the 2020 harvest.

### 2.4. Physicochemical Analysis

The process of fermentation activity was monitored daily by measuring the density (g/L) and temperature (°C). The standard oenological parameters of wines were performed according to EU Official Methods [21]. The ethanol concentration %($v/v$) in 2020 wines was estimated by ebulliometry (Anadil, Anadia, Portugal) and corrected if necessary by using a Baume hydrometer (0–10 scale) at 20 °C. In 2021, alcohol was measured by distillation followed by the aerometric method of OIV (International Organisation of Vine and Wine) [22]. The pH values were obtained by using a digital calibrated pH-meter (Thermo Scientific™, Orion Star™ A211, Beverly, MA, USA). The amount of free and total $SO_2$ in wines was determined by using a Sulfilyser measurement device (Laboratoires Dujardin-Salleron, Noizay, France). Total phenol index (TPI) was calculated according to the method by Ribéreau-Gayon et al. [23]. The reducing substances (comprising all the sugars) were determined according to the standard method of OIV [24]. In the 2021 harvest season, acetic acid was determined enzymatically in grapes with different degrees of dehydration and in juice using the RIDA®CUBE apparatus (R-Biopharm AG, Darmstadt, Germany). All analyses were carried out in duplicate. The RSDs (Relative Standard Deviation = (standard deviation/mean) × 100%) were below 10%.

### 2.5. Quantitative Determination of Mineral Content

The ICP-OES analytical technique (Inductively Coupled Plasma-Optical Emission Spectrometry) was applied for determining the elemental composition of wines using the Thermo Scientific™ iCAP™ 7400 ICP-OES analyser (Thermo Fisher Scientific, Bremen, Germany). The content of 11 elements (Na, K, Ca, Mg, P, S, Fe, Cu, Zn, Mn and B) in the wine samples was determined in accordance with the OIV criteria [25]. The RSDs (Relative Standard Deviation = (standard deviation/mean) × 100%) were below 5%.

### 2.6. Statistical Analysis

All statistical analyses were performed using R Statistical 3.6.1 Software ("Datamotus" LLC, Yerevan, Armenia) for Windows. The data was originally recorded in an M/Excel Worksheet and then imported into program R. Principal Component Analysis (PCA) was used to reduce a large amount of data into smaller ones with the function PCA (FactoMineR package) [26]. The interpretation/visualization of the PCA results was done using the R package "factoextra". All data visualizations were designed using the "ggplot2" and "ggthemes" packages. For hierarchical cluster analysis (HCA), the R package "dendextend" was applied.

## 3. Results and Discussion

### 3.1. Grape Ripening and Dehydration

The evolution of the ripening and further dehydration process is shown in Table 2 for the 2020 harvest. The drying process was stopped at 27.9 °Brix when grapes had lost 32.53% of their weight, as this is common practice in *passito*-style wines [6]. In 2021, grapes were harvested at 24.7 °Brix and dried under greenhouse conditions. The dehydrated juice before fermentation yielded 32.8 °Brix, pH 4.03 and 4.6 g/L (tartaric acid) total acidity.

**Table 2.** Muscat grape berry ripening and dehydration during the 2020 vintage.

| Phase | Date | Weight of 100 Berries (g) | Weight Loss (%) | Volume of Juice (mL) | Juice Yield (mL/g) | Brix (°Brix) | pH | Total Acidity (g/L Tartaric Acid) |
|---|---|---|---|---|---|---|---|---|
| Ripening | 3 August | 447.6 | - | 204 | 0.46 | 17.3 | 2.86 | 7.7 |
| | 10 August | 426.6 | - | 222 | 0.52 | 19.4 | 3.18 | 5.7 |
| | 17 August | 401.8 | - | 238 | 0.59 | 18.9 | 3.21 | 5.3 |
| | 24 August | 495.0 | - | 250 | 0.51 | 20.1 | 3.38 | 5.0 |
| | 31 August | 520.0 | - | 216 | 0.41 | 21.4 | 3.44 | 4.5 |
| Dehydration | 4 September | 570.6 | 0 | 278 | 0.49 | 24.7 | 3.95 | 4.1 |
| | 7 September | 438.8 | 23.1 | 133 | 0.30 | 29.4 | 3.87 | 4.9 |
| | 8 September | 469.7 | 17.7 | 138 | 0.29 | 29.7 | 3.91 | 3.9 |
| | 10 September | 385.0 | 32.5 | 128 | 0.33 | 27.9 | 3.75 | 4.5 |

### 3.2. Fermentation Kinetics

The juice extraction by foot without pressing (*prototropum*) only enabled the procurement of juice for one trial. The amount of juice for the other modalities was enough to obtain experimental duplicates. After juice extraction and starter inoculation, the kinetics of fermentations were monitored from the beginning to the end of alcoholic fermentation (Figure 2). The density decreased below 1000 g/L in the PW, SW, StW and WW samples. In the case of CW sample, obtained from dried grapes, stuck fermentation occurred due to the higher initial content of sugar in the juice. The semi-industrial scale fermentation performed in 2021 is shown in Figure 3. As in 2020, stuck fermentation occurred at a density of 1005 g/L.

### 3.3. Physicochemical Analysis

The results of physico-chemical parameters of the wines after fermentation are reported in Table 3. The highest ethanol concentration (17%, *v/v*) in 2020 was obtained with the wine WW, resulting from rehydrating grapes with one-year-old wine. The addition of boiled juice to the wine CW diluted the ethanol content to 12.6% (*v/v*) and increased the sugar concentration to 125.8 g/L. The final sugar concentration evidenced clear stuck fermentations for the WW and CW samples. Volatile acidity reached over 2 g/L (acetic acid) in the CW and CWC samples. The high value of the total phenol index (TPI) in CWC was explained by the addition of boiled juice.

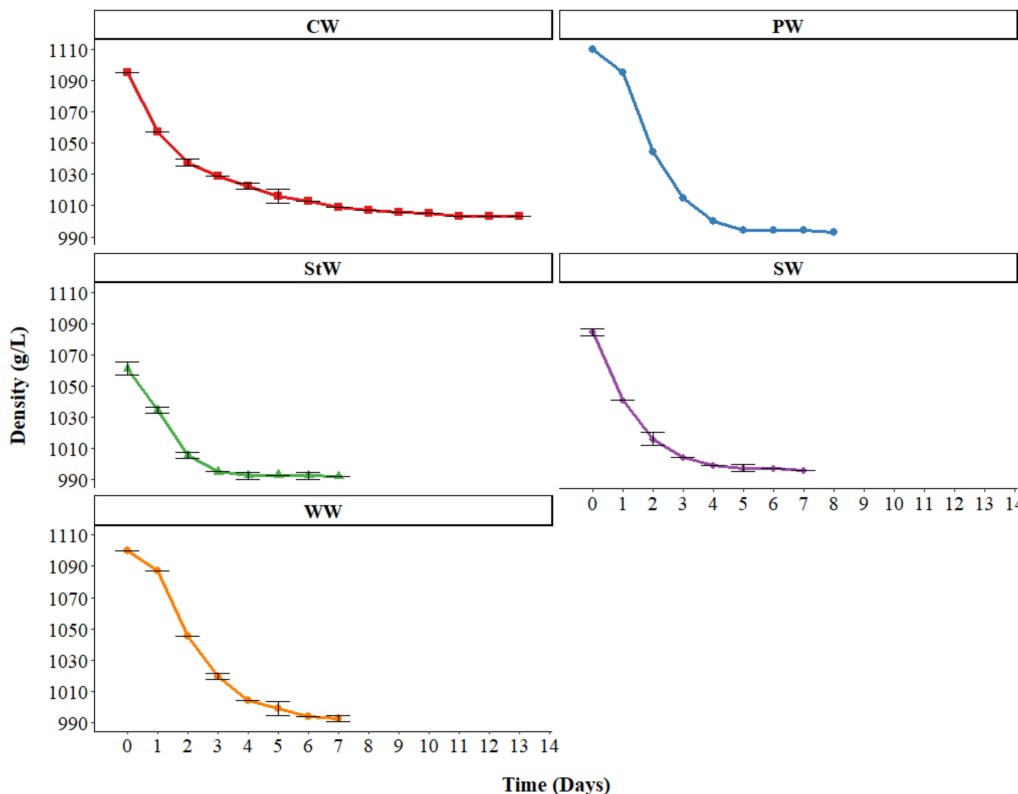

**Figure 2.** The fermentation kinetics of grape-musts in 2020 harvest (an average of two ferments except for PW; horizontal black bars indicate the standard deviation; see Table 1 for sample description).

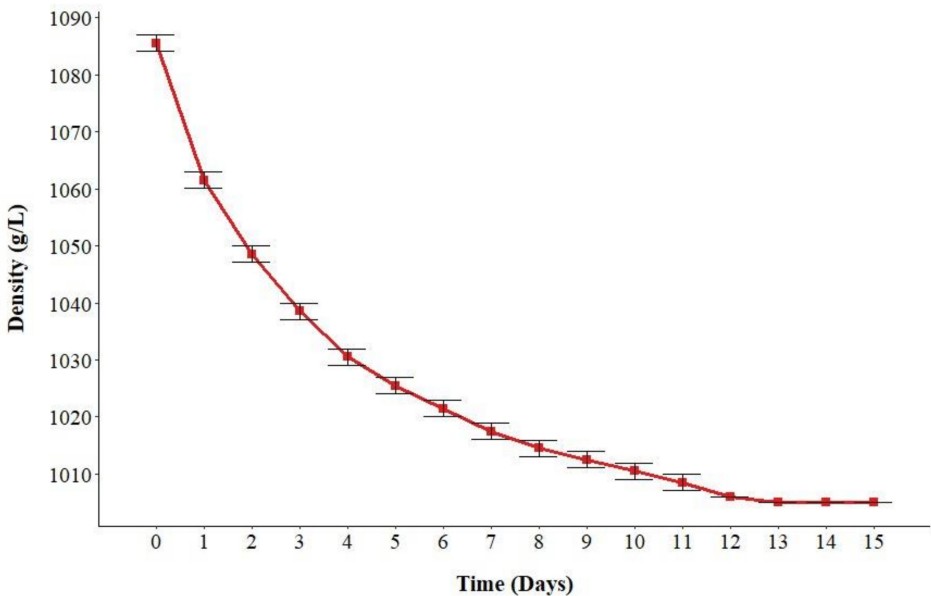

**Figure 3.** Semi-industrial scale wine fermentation in the harvest of 2021 (horizontal black bars represent the standard deviation of the mean of two repetitions).

**Table 3.** Wine physicochemical characterization.

| Wine | Alcohol at 20 °C (%, v/v) | Reducing Substances (g/L) | Total Acidity (g/L Tartaric Acid) | Volatile Acidity (g/L Acetic Acid) | pH | Total SO₂ a (mg/L) | Free SO₂ (mg/L) | Total SO₂ (mg/L) | Density at 20 °C (g/L) | Total Dry Extract (g/L) | Total Phenol Index |
|---|---|---|---|---|---|---|---|---|---|---|---|
| PW [b] | 15.0 | 0.8 | 5.1 | 0.21 | 3.90 | 28 | 43 | 138 | 989.8 | 27.9 | 7.2 |
| SW [c] | 14.8 ± 0.1 | 2.6 ± 0.4 | 4.7 ± 0.0 | 0.50 ± 0.01 | 3.85 ± 0.02 | 20 ± 0 | 19 ± 1 | 125 ± 0 | 994.2 ± 0.0 | 38.8 ± 0.0 | 7.7 ± 0.0 |
| StW [c] | 14.8 ± 0.0 | 0.7 ± 0.1 | 5.3 ± 0.1 | 0.32 ± 0.01 | 3.59 ± 0.03 | 33 ± 0 | 41 ± 0 | 157 ± 2 | 990.0 ± 0.0 | 27.9 ± 0.0 | 8.8 ± 0.0 |
| WW [c] | 17.0 ± 0.0 | 6.1 ± 0.0 | 4.7 ± 0.3 | 0.39 ± 0.02 | 3.95 ± 0.01 | 40 ± 0 | 30 ± 0 | 148 ± 4 | 990.4 ± 0.0 | 35.2 ± 0.0 | 8.6 ± 0.0 |
| CW [c] | 16.3 ± 0.1 | 17.9 ± 1.1 | 7.7 ± 0.3 | 2.26 ± 0.06 | 3.82 ± 0.01 | 13 ± 0 | 14 ± 1 | 72 ± 5 | 1000.2 ± 0.0 | 58.5 ± 0.0 | 8.7 ± 0.0 |
| CWC [c] | 12.6 ± 0.3 | 125.8 ± 1.8 | 8.0 ± 0.1 | 2.15 ± 0.06 | 3.94 ± 0.01 | - | 26 ± 1 | 144 ± 2 | 1048.0 ± 0.0 | 172.7 ± 0.8 | 40.2 ± 0.0 |
| CW21 [c] | 17.6 ± 0.3 | 36.0 ± 0.2 | 5.1 ± 0.1 | 1.16 ± 0.10 | 4.36 ± 0.03 | - | 12 ± 1 | 55 ± 6 | 1.0 ± 0.0 | 68.0 ± 0.5 | 12.0 ± 0.7 |
| CW21L [b] | 18.9 | 18.1 | 5.3 | 0.83 | 4.24 | - | 14 | 36 | 1.0 | 50.7 | 12.4 |

[a] Total sulphur dioxide concentration at the end of fermentation. [b] One trial. [c] Mean values ± standard deviations (S.D.) of duplicate trials.

The values of total sulphur dioxide at the end of fermentation reflect production by the starter yeast, probably stimulated by the high initial sugar content [27]. The high volatile acidity could be related to the response of *S. cerevisiae* to high initial sugar concentration, as also reported by Morales et al. [28], or by partially aerobic conditions [29], likely to occur in low-volume ferments. Indeed, the problem of high volatile acidity is common in sweet wines obtained from dried grapes [30]. The 2021 trial showed wines with lower levels, although close to the legal limit of 1.2 g/L acetic acid. This high value may be partially explained by the acetic acid in the juice (0.6 g/L) and not by the activity of acetic acid bacteria. Moreover, grape dehydration induced an increase in acetic acid of up to 0.25 g/L in a random sample of single berries. Probably, acetic acid increased in a mode similar to gluconic acid in gradually more dehydrated berries [31], thus justifying further research.

The high ethanol contents in 2021 were remarkable, showing an unexpected yeast performance, particularly with the wine obtained from the gross juice lees (5 L). Ethanol content reached 18.9% (*v/v*), much higher than the maximum level of 13% (*v/v*) [32] or 13.7% (*v/v*) [30]. This higher ethanol yield may be explained by the likely higher lipid concentration in the lees juice [33]. Concerning wine yields, 100 L of *passito*-style wine was obtained, resulting in 0.37 L/kg of dehydrated grapes, comparable to the yields reported by Rolle et al. [31].

A Principal Component Analysis (PCA) was carried out on the whole data set considering the sulphite concentrations after addition. The first two principal components explained 84% of the total variability of the entire dataset in PCs of 100% (Figure 4). The PC1 axis yielded the highest variation (59.7%), whereas the PC2 axis yielded the lowest (24.3%). The PCA indicates that SW, PW, StW and WW samples were highly positively

correlated with PC1 in terms of free and total sulphur dioxide, but showed a weak positive correlation with alcohol at 20 °C. The CWC sample has large negative loading on PC2, as expected from the lower ethanol and higher reducing substances, total dry extract and density at 20 °C.

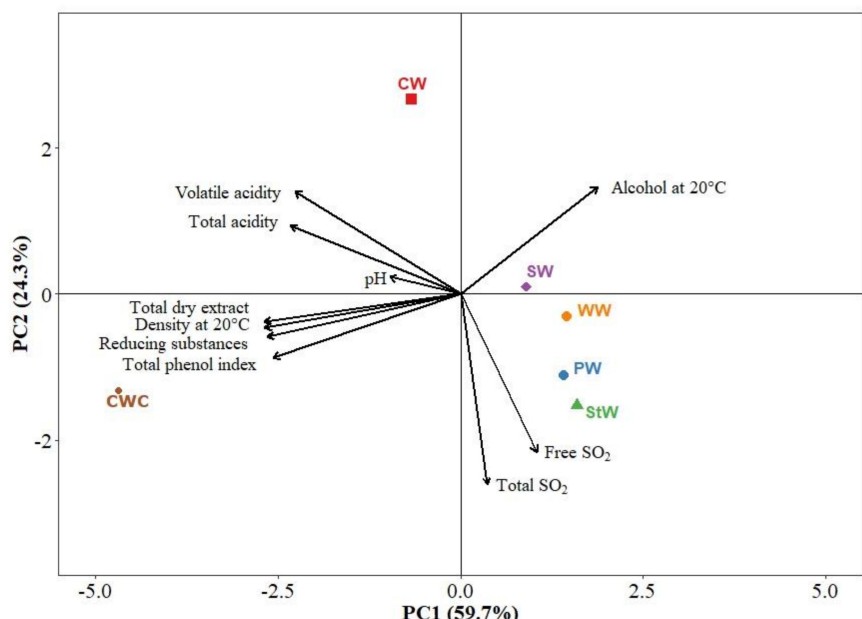

**Figure 4.** Bi-plot of the Principal Component Analysis (PCA) with the main physico-chemical parameters of wines (sample description in Table 1).

The analysis of the wines obtained after dilution of pomace with tartaric acid solution is shown in Table 4. The fermentation was completed with reducing substances below 2 g/L and volatile acidity below 0.8 g/L. The ethanol level reflected the different pomace dilutions ranging from 10.2 to 16.0% (*v/v*).

**Table 4.** Physicochemical analysis of *piquettes*.

| Sample | Alcohol at 20 °C (%, *v/v*) | Reducing Substances (g/L) | Total Acidity (g/L Tartaric Acid) | Volatile Acidity (g/L Acetic Acid) | pH | Free SO$_2$ (mg/L) | Total SO$_2$ (mg/L) |
|---|---|---|---|---|---|---|---|
| PSW | 14.1 | 1.4 | 4.95 | 0.74 | 3.87 | 43 | 170 |
| PWW | 16.0 | 1.0 | 4.35 | 0.62 | 3.97 | 48 | 160 |
| PStW | 10.2 | 0.1 | 4.50 | 0.45 | 3.75 | 48 | 160 |
| PCW | 11.5 | 0.9 | 5.70 | 0.75 | 3.62 | 66 | 158 |

### 3.4. Mineral Analysis

The mineral analysis of 6 elements (Na, K, Fe, Cu, Zn and B) was performed, ensuring to check: (a) the influence of salt addition during rehydration in Na levels likely to overcome legal limits; (b) the concentrations of Cu, Fe, Zn and B, subjected to maximum legal limits [34]; and (c) the concentration of K as a function of grape maceration. The mineral concentrations in wines, beverages and *piquettes* (in mg/L) are presented in the Supplementary Table S1. As expected, the highest value of Na was found in the SW sample, with an average concentration of 1331.4 mg/L, while in the other wines it was below 80 mg/L. The addition of boiled juice was also reflected in the highest content of K in the CWC wine. The composition of the *piquettes* followed the trend that would be expected from the respective pomaces of origin.

The concentrations of Cu, Fe, Zn and B were below the respective maximum legal levels (1 mg/L for Cu, 20 mg/L for Fe, 5 mg/L for Zn and 80 mg/L as boric acid) [34]. As expected, the wine with added saltwater was clearly over the limit for Na (80 mg/L). The

mineral contents were within the range found in Greek home-made wines [35], except for the Na and K contents. The higher K concentration in this study may be explained by the likely effect of pre-fermentative maceration [36,37].

## 4. Conclusions

This study represents the first step in the adaptation of ancient techniques to produce wines from dehydrated grapes. Some modalities resemble the *passito*-style wine, but others require adaptation to legislation, mainly in the case inducing Na levels over the maximum legal limit. Nevertheless, the results showed the feasibility of adapting ancient Roman protocols to produce wines that have acceptable physicochemical quality. The main difficulty arose from the fermentation of high sugar juices that induced a volatile acidity higher than the permitted value (1.2 g/L acetic acid). The semi-industrial trial enabled the reduction of volatile acidity, although it was still near the legal limit. The pomace dilution with tartaric acid solution enabled the procurement of a product (*água-pé* or *piquette*) with potential market interest, once permitted by legislation. In this way, it will be possible to increase the volume of the resulting products to compensate the low wine yield from the dehydrated grapes. Future work will assess consumers' acceptance of the beverages produced during this study.

**Supplementary Materials:** The following supporting information can be downloaded at: https://www.mdpi.com/article/10.3390/fermentation8020085/s1, Table S1: The concentration levels of essential macro- and microelements in wines and beverages (in mg/L).

**Author Contributions:** M.H. and R.V. performed the 2020 fermentations trials; M.H. performed the 2021 fermentation trial and colour illustration; J.G.-S. carried out wine physico-chemical analysis; M.M. and H.R. were responsible for the mineral analysis; M.M.-F. designed the concept of the study; M.H. and M.M.-F. wrote the manuscript. All authors have read and agreed to the published version of the manuscript.

**Funding:** This work was conducted under the financial support by the "Armenian Communities Department" of Fundação Calouste Gulbenkian within the scope of the Short-Term Grant for Armenian Studies (N°239337/2020, https://gulbenkian.pt/en/bolsas-lista/short-term-grant-for-armenian-studies-2022/, accessed on 30 December 2021); and by national funds through FCT—Fundação para a Ciência e a Tecnologia, I.P., in the scope of the project Linking Landscape, Environment, Agriculture and Food Research Centre (Ref. UIDB/04129/2020 and UIDP/04129/2020). Mkrtich Harutyunyan (M.H.) was granted a Ph.D. scholarship from the University of Lisbon (2018–2021).

**Institutional Review Board Statement:** Not applicable.

**Informed Consent Statement:** Not applicable.

**Conflicts of Interest:** The authors declare no conflict of interest.

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
