# Peer review of "Adaptation of Ancient Techniques to Recreate ‘Wines’ and ‘Beverages’ Using Withered Grapes of Muscat of Alexandria"

_fermentation, doi:10.3390/fermentation8020085_

Round 1

Reviewer 1 Report

As requested, this study was conducted based on the traditional methods of wine and piquettes made from dehydrated grapes of Muscat of Alexandria. The physicochemical analysis showed that it was possible to recreate ancient winemaking techniques. However, this manuscript lack of novelty, and the conclusions need carefully organized, more relative references should be provided. Therefore, the manuscript need major revision is needed. Specific comments are listed:

In Materials and Methods:

  1. More specifications should be clarified. Line 65 "regularly turned over..." the definition of "regularly" are not clear? The specific time should be marked in the manuscript.

Line103, the  number of sensorial panelists is not enough to collect meaningful data. Besides, gender and age of panelists are not well explained. In Results and discussion:

  1. Figures in the paper need to modified with error bars, and data in the tables should be represented as "mean ± standard deviation".
  2. In Table 3, why only one PM group is one trial? This is different from other processing groups, please explain. PW treatment only done with one trial? It is so confused why the treatments couldn’t have done with three replicates.
  3. In 3.4 explanation of mineral Analysis is confused. Author have to provide clear and concise conclusions, more relative references need to supplied in this manuscript. It is suggested to explain the relationship between the research content in this section and this article.

Author Response

As requested, this study was conducted based on the traditional methods of wine and piquettes made from dehydrated grapes of Muscat of Alexandria. The physicochemical analysis showed that it was possible to recreate ancient winemaking techniques. However, this manuscript lack novelty and the conclusions need carefully organized, more relative references should be provided. Therefore, the manuscript needs major revision is needed.

- We inserted more references to support the study approach and the obtained results. The readers are also directed to a review made by the authors on ancient winemaking techniques and wine styles that is in press in the journal Beverages.

Specific comments are listed below:

In Materials and Methods:

  1. More specifications should be clarified. Line 65 "regularly turned over..." the definition of "regularly" are not clear? The specific time should be marked in the manuscript.

- We specified it better in lines 81-82, hopefully.

  1. Line103, the number of sensorial panellists is not enough to collect meaningful data. Besides, gender and age of panelists are not well explained.

- The purpose was not to make a sensory analysis but just an off-flavour screening. The sensory analysis with more than 100 tasters was done and will be submitted in a next article. Therefore, we decided to remove to part of sensory analysis to avoid confusion.

  1. In Results and discussion: Figures in the paper need to modified with error bars, and data in the tables should be represented as "mean ± standard deviation".

- Corrected accordingly.

  1. In Table 3, why only one PM group is one trial? This is different from other processing groups, please explain. PW treatment only done with one trial? It is so confused why the treatments couldn’t have done with three replicates.

- The volume of wine issuing from dried grapes was a limitation for the repetitions. The PM volume was little and we decided to make one trial with the same volume as the other trials instead of doing two trials with half of the volume. Different fermenting volumes also influence oxygen availability and fermentation performance. The similarity in duplicates is an indicator of the reproducibility of the fermenting conditions.

  1. In 3.4 explanation of mineral Analysis is confused. Author have to provide clear and concise conclusions, more relative references need to supplied in this manuscript. It is suggested to explain the relationship between the research content in this section and this article.

- The sodium concentration was mandatory in order to check the effect of salt addition and its relation with the legal limits. The full mineral analysis would not be necessary, indeed, but the method gives all parameters simultaneously. We moved the data to a supplementary table to minimize confusion.

Reviewer 2 Report

  1. The Abstract section should be a concise summary of the entire paper and should include a clear description of what the author intends, the importance and relevance of the study, a brief description of the methods, the most important result and the interpretation given to the key result.
  2. This research is not justified: the products obtained are not considered wines by the OIV regulations in force.
  3. The degree of innovation and novelty is not presented.
  4. Why is it necessary to determine some minerals from the products obtained? Color analysis of phenolic, volatile compounds and organoleptic analysis would have been very useful.
  5. It is not possible to compare samples obtained in different years, starting from different raw materials, using different treatments. There are many variables that were not taken into account and the results are not representative.
  6. L22-33: What was the semi industrial trail to reduce the volatile acidity? Was it an ancient technique? Anyway it is not allowed by legislation.
  7. L 24: Mousiness in wine has another cause!!!
  8. L 37-38: Normally, passito wines have less alcohol but high residual sugar.
  9. L 67-69: 30 Brix is not corresponding to 30% weight loss. Please reformulate in a scientific way.
  10. L 72-76: Please reformulate. In ancient time, wine was diluted with salty water. Please add references for the technique you describe.
  11. L 78-88: Please reformulate. There is confusion on the experiment. Also look on SO2 treatment. Confusing too.
  12. L 92: Ebuliometry is only used for dry wines.
  13. L136: Table 2: In 2020 between 17 to 31 Aug, you have an increase of weight of 100 berries (g) from 401.8 g to 520.0 g (about 130%) and a volume of juice from 238 mL to 216 ml. Please explain.

Author Response

  1. The Abstract section should be a concise summary of the entire paper and should include a clear description of what the author intends, the importance and relevance of the study, a brief description of the methods, the most important result and the interpretation given to the key result.

- We hope to have rephrased the abstract accordingly.

  1. This research is not justified: the products obtained are not considered wines by the OIV regulations in force.

- The modalities PW, CW and CWC are passito-style wines because correspond to the fermentation of juices originating from dried grapes only. StW and WW include rehydration with one year old wine that may be compared to the governo all’uso toscano where withered grapes are rehydrated in fresh new wine (DOI 10.1111/ajgw.12434). The addition of salt is not allowed but that was clearly stated in the text so that readers are aware of this limitation. Piquettes may be home consumed but today there are already some commercial products (see for instance https://www.jancisrobinson.com/articles/piquette-summer-wine-everyone) that justify a scientific exploration. Therefore, although some products are not considered wines by the OIV, research on their properties anticipates possible future developments.

  1. The degree of innovation and novelty is not presented.

- The novelty of this study concerns the adaptation of roman descriptions that in some cases have present counterparts (e.g, passito) but in others have not, such as the rehydration with wine or salt water. As far as we are aware, piquette´s experimentation is a novelty in research. Lines 66 to 72 address this question.

  1. Why is it necessary to determine some minerals from the products obtained? Color analysis of phenolic, volatile compounds and organoleptic analysis would have been very useful.

- The analysis of minerals is justified by the legislation. Indeed, we have a whole range of data involving volatile GC analysis, color (CIELAB) and organoleptic analysis but that is to be presented in a sequential paper.

  1. It is not possible to compare samples obtained in different years, starting from different raw materials, using different treatments. There are many variables that were not taken into account and the results are not representative.

- The purpose was not to compare different years but to make a trial closer to industrial conditions, mainly having in mind the requirement for lower volatile acidity.

  1. L22-33: What was the semi industrial trail to reduce the volatile acidity? Was it an ancient technique? Anyway it is not allowed by legislation.

- The 2021 trial was done in passito style. So the process is legal, as far as we are aware. The purpose of this trial was to show that volatile acidity could be lower. Indeed, the volatile acidity was close to other reports in this kind of fermentations.

  1. L 24: Mousiness in wine has another cause!!!

- We know 3 possible causes: Brettanomyces, lactic acid bacteria, chemical oxidation. In our case we speculate that the cause should be the third given the absence of added sulphite and unlikely activity by those microbes. However, we deleted these observations because we will address sensory analysis in a next article.

  1. L 37-38: Normally, passito wines have less alcohol but high residual sugar.

- Thank you for the correction. We introduced references that reports a wider range of ethanol and sugar levels (lines 37-40).

  1. L 67-69: 30°Brix is not corresponding to 30% weight loss. Please reformulate in a scientific way.

- We changed the phrase to clarify the concept, clearly Brix is not related to weight loss.

  1. L 72-76: Please reformulate. In ancient time, wine was diluted with salty water. Please add references for the technique you describe.

- We added references to support the statements (lines 63 a 67).

  1. L 78-88: Please reformulate. There is confusion about the experiment. Also, look at SO2 treatment. Confusing too.

- Changed in accordance.

  1. L 92: Ebulliometry is only used for dry wines.

- Ebulliometry was used in 2020 with necessary corrections for sugar concentration. This method had to be used given the low volume of sample required. In 2021 we used distillation but this was not described by mistake. We are grateful for this comment and the method description was now corrected accordingly (lines 120-123).

  1. L136: Table 2: In 2020 between 17 to 31 Aug, you have an increase of weight of 100 berries (g) from 401.8 g to 520.0 g (about 130%) and a volume of juice from 238 mL to 216 ml. Please explain.

- This is a result according to our experience with Muscat grapes. During sampling to obtain juice, the extraction by hand squeezing becomes less efficient and juice yield decreases. However, this methodological limitation is not relevant because the important is to follow Brix increase to determine when to harvest.

Round 2

Reviewer 1 Report

According to the requirements, this paper reproduces the old methods of wine and pigot made from dehydrated grapes. The revised version has corrected many problems, In General, as a kind of recreative wines, its novelty is admitted, and this research is also very interesting. Therefore, the manuscript need minor revision is needed. Here are some suggestions:

  1. The revised manuscript does not carry out sensory analysis experiments, but line 218 mentioned the results of sensory analysis in results and discussion.
  2. The raw materials harvested from 2020 must be different from those harvested in 2021. How can the author eliminate these differences in raw materials?
  3. The conclusions obtained from line 222-224 have no corresponding reference support. They are subjective assumptions with low reliability. It is suggested that the author analyze them according to relevant references.
  4. What is the meaning of adding Figure 5 to the manuscript? There is no conclusion about the research content drawn from Figure 5 in the manuscript.
  5. The analysis of minerals in the article is less relative to this article. It suggested that the manuscript only presented the relevant charts of key mineral analysis. In addition, the article should clearly indicate the necessity of minerals analysis of contribution.

Author Response

The revised manuscript does not carry out sensory analysis experiments, but line 218 mentioned the results of sensory analysis in results and discussion.

- This was a mistake that now was properly corrected by removing the respective text and corresponding references.

The raw materials harvested from 2020 must be different from those harvested in 2021. How can the author eliminate these differences in raw materials?

- We can not eliminate differences because grapes ripen differently according to the different harvest years. The differences should have been minimized because (a) the grapes came from the same experimental vineyard where all production was directed to the trials presented in this article, and (b) viticulture practices were similar in both years. This explanation was included in Material in Methods (section 2.1). Indeed, differences in raw materials are impossible to circumvent when grapes from two different years were used.

The conclusions obtained from line 222-224 have no corresponding reference support. They are subjective assumptions with low reliability. It is suggested that the author analyze them according to relevant references.

- We think that lines 222-224 refer to the questions of high ethanol yields in grape juice before fermentation or the acetic acid levels. Three references were added to support the observations (31, 32 and 33). Concerning acetic acid in juices previous to fermentation, we could not find results in the literature.

What is the meaning of adding Figure 5 to the manuscript? There is no conclusion about the research content drawn from Figure 5 in the manuscript.

- This figure and corresponding text were removed, indeed the conclusions related to sample chemical differences were already drawn from figure 4.

The analysis of minerals in the article is less relative to this article. It suggested that the manuscript only presented the relevant charts of key mineral analysis. In addition, the article should clearly indicate the necessity of minerals analysis of contribution.

- The table related to mineral composition (Supplementary data) was reduced to the concentrations of relevant minerals. The text was corrected accordingly and a clear explanation of the necessity of this analysis was introduced.  

Reviewer 2 Report

We recommend rejecting the paper in this form, due to the lack of scientific accuracy and the representativeness of the results.

Author Response

We are aware that the trials should have been performed at least in triplicates. To obtain these triplicate samples, the volume of each trial would be reduced thus increasing the differences from realistic winery conditions. Moreover, all Muscat grapes from the experimental vineyard were used in this research. For these reasons, we decided to run duplicate experiments.

In the case of the prototropum, the juice volume did not allow us to perform duplicates but we decided to present the results given the significance of this wine in Roman texts where it is regarded as the most expensive wine.

These methodological options were also required to obtain enough wine to perform consumer tastings with more than 100 individuals that will be the subject of a sequent article.